# Achieving a Sustainable Transportation System via Economic, Environmental, and Social Optimization: A Comprehensive AHP-DEA Approach from the Waste Transportation Sector

Hala Hmamed [1,*], Asmaa Benghabrit [2], Anass Cherrafi [3] and Nadia Hamani [4]

1   LM2I Laboratory, ENSAM, Moulay Ismail University, Meknes 50500, Morocco
2   LMAID Laboratory, ENSMR, Rabat 10000, Morocco; benghabrit@enim.ac.ma
3   LAPSSII Laboratory, EST-Safi, Cadi Ayyad University, Safi 46000, Morocco; a.cherrafi@uca.ac.ma
4   LTI Laboratory, University of Picardie Jules Verne, 02100 Saint Quentin, France; nadia.hamani@u-picardie.fr
*   Correspondence: hala.hmamed@edu.umi.ac.ma

**Abstract:** Given the growing global emphasis on sustainable transportation systems, this research presents a comprehensive approach to achieving economic, social, and environmental efficiency in transport within the waste management sector. To address the different challenges of sustainable transportation issues, this paper presents a hybrid multi-criteria decision-making (MCDM) approach that incorporates the analytic hierarchy process (AHP) along with data envelopment analysis (DEA) for sustainable route selection. By leveraging the strengths of both methods, this approach reconciles conflicting requirements and diverse perspectives, facilitating effective decision making. This paper involves identifying relevant criteria for route evaluation, engaging waste management company experts and stakeholders in pairwise comparisons using AHP. Furthermore, DEA is used to calculate route efficiency based on the inputs and outputs of the system. These evaluations enable the identification of the most effective and sustainable routes. This proposed methodology empowers decision makers and transportation policymakers to develop an effective decision-making tool for addressing waste transportation challenges in developing countries. The study contributes to the growing body of research on sustainable waste management practices and provides insights for waste management companies and decision makers on how to optimize waste transportation routes while reducing economic, social, and environmental impacts.

**Keywords:** sustainable transport; multiple criteria decision-making approach; analytic hierarchy process; data envelopment analysis; route selection; waste transport

## 1. Introduction

Transportation systems have significant environmental impacts due to their significant contribution to air pollution [1]. The increasing reliance on vehicles for daily needs presents various challenges, including climate change, environmental degradation, and health-related issues [2]. In fact, the harmful pollutants generated by the transportation sector have the potential to severely affect global health [3]. Therefore, it becomes crucial to develop a sustainable transportation infrastructure that prioritizes the efficient use of energy [4]. In this context, communities face the challenge of promoting sustainable transportation solutions to mitigate the different effects of pollution and excessive energy consumption [5]. Moreover, the growing population and business expansion bring new additional challenges, emphasizing the importance of sustainable transportation in meeting both social and economic requirements while sustaining the environment [6]. As a result, adopting sustainable transportation practices has the positive effect of aligning current and future economic development, enhancing transportation efficiency, and preserving the environment [7].

According to [8], sustainable transportation refers to a transportation system that reduces the social and environmental impacts. In fact, sustainable transportation solutions are needed to address the increasing requirement for transportation in expanding urban areas while preventing potential unfavorable social, economic, and environmental consequences [9]. Transportation planning, among the most challenging issues faced by urban transportation systems [10], frequently results in high costs, unscheduled delays, increased energy consumption, and increased emissions, pollution, and noise [11,12]. As a result, it has enormous environmental, economic, and social consequences. In addition, transportation planning in cities, particularly in developing countries, remains a serious concern due to growing urbanization [13]. It raises transportation costs by increasing fuel consumption, carbon emissions, and distribution inefficiencies, resulting in significant consequences for the environment and human health [14,15].

Given the number of conflicting economic, social, and environmental objectives involving sustainable transportation solutions, this issue can be classified as a multiple-criteria decision-making (MCDM) problem in which all of these objectives must be considered concurrently [16–18]. In current sustainable transportation problems such as route selection, the focus is not to achieve the lowest total price or the shortest delay [19]. Instead, the objective is to discover an optimal approach that maximizes beneficial impacts while also taking non-benefit characteristics into account [20]. As a result, decision makers are ultimately faced with a challenging set of parameters relative to the available alternative options.

Numerous studies have examined a broad range of variables from different perspectives in the field of route selection decision making [21,22]. These factors frequently include transportation cost effectiveness, time, capacity, and distance optimization. Previous works focused on environmental factors including fuel consumption, pollution, waste, and emissions [23–25]. These criteria have been commonly explored using MCDM techniques, allowing researchers to prioritize alternatives and suggest the most optimal choice based on their models. For example, the analytic hierarchy process (AHP) has different applications in the area of transportation [26]. For instance, AHP was used by [27] to determine the optimal logistics network for a crucial freight route. Moreover, [15] used AHP to select the best solutions for developing an environmentally friendly transportation system. In addition, route effectiveness is a critical factor in sustainable transportation. It is determined by several important variables that vary across various transportation network routes and regions.

The performance of transportation networks has frequently been a source of concern and requires more assessment [28]. In this context, the data envelopment analysis (DEA) approach may be performed to evaluate transport route efficiency given a particular set of inputs and outputs in the field of transportation [29]. It is extensively used in manufacturing, operation, leadership, and economics to experimentally evaluate operational effectiveness [30]. The DEA has a broad range of applications for transportation-related issues. For example, [31] presented an approach to employing DEA in maximizing the efficiency of metropolitan public transportation networks. Furthermore, [32] used DEA to assess the effectiveness of possible various state transportation agencies in roadway management. The authors of [33] developed a framework to examine the operational effectiveness of transportation networks. In addition, [34] presented a solution to the logistics challenge that takes into account various inputs and outputs associated with each transportation route. Other research proposed a multimodal transportation routing approach based on the cost and emissions criteria [35].

Nonetheless, using a single MCDM technique may not provide the most accurate results when compared to employing an integrated approach that combines multiple MCDM methods [36,37]. In this paper, we used an integrated approach for AHP-DEA for waste transportation route selection. The integration of these two techniques proves to be an effective strategy, especially when dealing with decision-making challenges in real case studies [37,38]. By doing so, the limitations of individual techniques can be mitigated while their respective advantages can be complemented, leading to more robust

and reliable results. Several previous studies used the integrated AHP-DEA approach to address transportation optimization concerns from various perspectives including road networks [39], supply chains [40], and urban intermodal systems [41]. The weights obtained from AHP are then incorporated into DEA to evaluate the efficiency of different routes in terms of resource utilization and the logistics transportation process. Researchers have used AHP-DEA to optimize freight transportation routes considering different freight parameters [42–45]. In this context, criteria related to energy efficiency, traffic affordability, and costs are often included. In addition, AHP and DEA have been applied to design and optimize public transportation systems. For instance, [44] used the same approach to evaluate the performance of the main public road transport organizations, AHP was used for identifying the most important criteria, and DEA was subsequently employed for the evaluation of the efficiency of different transport roads in meeting the identified criteria and objectives. Additionally, AHP and DEA have been integrated into multimodal freight transportation planning, where various transportation modes (e.g., road, rail, air, and water) are considered [46,47].

Although multiple examples of hybrid approaches regarding multiple perspectives have often been addressed, none have provided a method for identifying the most optimal route that comprehensively considers sustainable requirements along with stockholders' and decision makers' preferences via a combination of two MCDM approaches, AHP and DEA. Previous works have used combined AHP and DEA for transportation issues, but none of them addressed sustainable transportation systems. Furthermore, this research uses a real case study from the waste transportation sector in a developing North African country. The study was conducted in a medium-sized city with an estimated population of approximately 632,079 individuals covering an area of around 550 square kilometers and a population density of approximately 1149 people per square kilometer according to the Moroccan High Commission for Planning and Demographic Statistics.

Currently, the waste management sector is one of the main concerns that impact environmental, social, and economic aspects [48]. As a result, sustainable development has been recognized as a critical requirement that the waste management industry should take into consideration [49]. Although, all waste management operations and processes have a direct influence on the environment [50]. This study specifically addresses the waste transportation aspect, i.e., transportation of the collected waste from various collection points in different regions within small–medium size cities to the disposal site. Although waste management research has extensively covered aspects like waste generation, classification, tracking, and effectiveness assessment [13,51], the topic of waste transportation has been largely neglected and received limited attention, particularly in developing countries where traditional waste management approaches are still dominant.

Consequently, there is a requirement to expand research on transportation challenges to integrate sustainability concerns. In this research, we provide a holistic approach to address transportation route selection issues in the waste transportation sector by combining AHP and DEA techniques. In fact, transportation route selection challenges require optimizing multiple goals including minimizing costs [52], maximizing efficiency scores [53], decreasing the environmental impacts of the ranked routes [54], and minimizing emissions [55]. In this context, AHP is used to rank the possible routes based on sustainable characteristics involving societal, financial, and environmental. The AHP-derived scores are employed to determine the significance of each criterion via the insights of experts and stakeholders. Subsequently, the DEA methodology integrates multiple inputs and outputs within the waste transportation context, enabling the generation of effectiveness assessments for each waste transportation route. The resulting scores are then employed to determine the optimal route selection for achieving a sustainable transportation system.

Overall, this study provides three significant contributions. First, it presents a hybrid MCDM methodology that combines AHP and DEA. This combination has not been previously explored in the context of transportation route selection. AHP is employed to quantify decision makers' preferences via the assignment of relative weights, while

DEA is used to generate route scores based on comprehensive criteria. The integrated AHP-DEA results facilitate the evaluation of alternative routes based on their efficiency scores. Second, this study incorporates social, economic, and environmental factors into the criteria for selecting sustainable routes. This comprehensive approach enables a holistic assessment of route options. Finally, despite the substantial impact of waste transportation on urban societies, prior researches have not addressed the area of waste transportation route selection. Consequently, this research bridges a critical gap by addressing a real case scenario involving a waste transportation company in a developing country. Furthermore, a sensitivity analysis was conducted to validate the robustness of the results by adjusting the criteria prioritization within this approach.

This paper is organized as follows. Section 2 discusses sustainable transportation challenges and the MCDM techniques. The methods of analysis, methodology, and parameter settings of the AHP and DEA approach are explained in Section 3. The results, sensitive analysis, and findings are summarized in Section 4. The following part presents a discussion of theoretical and managerial implications, limitations, and future works. The final section provides the conclusion of the paper.

## 2. Literature Review

Sustainable transport has become an important element in structuring urban operations [56]. In this context, sustainable transportation is a key factor to tackle environmental, social, and economic challenges [57]. Eventually, the incorporation of sustainable solutions into transportation networks has grown significantly. Such growth is strictly related to globalization and urban development due to the large increase in transport needs [2]. The rise in interest in sustainable transportation can be related to community initiatives that emphasize a higher focus on living standards and the efficient administration of public assets [58]. As a result, developing a sustainable transportation system is vital to address significant major problems including climate change, air pollution, traffic congestion, and resource depletion [5].

In fact, traditional transportation planning is primarily concerned with expanding availabilities to meet economic issues. Nonetheless, it overlooked social and environmental challenges such as energy consumption, noise pollution, emissions, energy consumption, ecological damage, waste generation, and road safety [58,59]. Consequently, effective transportation planning which includes route selection is a crucial contributing component in establishing sustainable transportation systems [54]. Indeed, the classification of various transportation routes based on sustainable considerations assists in evaluating their significance in achieving established objectives [28]. However, economic, environmental, and social factors may not exclude the selection of other vital components for transportation route planning. In other words, current transportation planning must be achieved by balancing sustainable aspects and transportation features.

Sustainable transportation planning is known as an integrated approach to elaborate effective, balanced, and environmentally responsible transportation systems [10]. It entails integrating numerous transportation variables such as vehicle, route type, distance, and time. To develop connected, accessible, and safe transportation networks, the planning process considers various aspects such as organizational and environmental elements, urban design, infrastructure, and technological development. It also addresses the requirements of different communities and nations [11]. Sustainable transportation planning promotes the use of renewable energy sources and cutting-edge technologies to maximize energy efficiency and reduce the overall environmental impacts [40]. Sustainable transportation planning seeks to build flexible and resilient transportation systems that support economic growth, enhance public health, and contribute to a more sustainable urban environment. This is achieved by promoting collaboration among stakeholders and engaging with public regulations [59,60].

In this context, previous researchers have studied the effectiveness of various public transportation systems, including cars, buses, trains, light rail, and innovative mobility solu-

tions like ride-sharing and bike-sharing programs [61]. Many researchers have investigated the economic and environmental benefits and challenges associated with different transportation techniques and solutions [53]. MCDM techniques have been extensively applied to address transportation planning challenges and find optimal solutions that consider multiple criteria and objectives. Indeed, MCDM approaches help transportation administrators in making informed decisions by considering several factors [36]. These factors include the studies of transportation mode selection [62], route prioritization [54], infrastructure investment [12], public transport planning [58], transport policy evaluation [46], and freight urban planning [45].

To handle transportation-planning challenges, AHP is frequently used when combined with other MCDM techniques such as DEA, TOPSIS (technique for order preference by similarity to ideal solution), ELECTRE (elimination and choice translating reality), VIKOR, and GRA (grey relational analysis). For example, Ref. [63] used integrated AHP with TOPSIS to address public transportation regarding bus selection. The study proposed a framework based on the hybrid technique to optimize the bus design with respect to a set of decision variables. Additionally, Ref. [64] employed AHP combined with VIKOR to determine efficient petrol station selection in transportation sectors. Moreover, Ref. [65] used an integrated approach of AHP and ELECTRE approach for sustainable path selection to determine transport-effective alternative solutions. Ultimately, these methods help decision makers to analyze complex trade-offs between conflicting objectives and arrive at informed and robust decisions for more optimized transportation systems [66]. In addition, the integration of methodologies into a hybrid evaluation of decisions is not a novel concept; it has been frequently recommended to develop an effective strategy by leveraging their strengths and complementing their limitations [67]. There are several papers on hybrid MCDM techniques for transportation systems as shown in Table 1.

**Table 1.** Existing studies on transportation MCDM combined approaches.

| Reference | Research Context | Applied Approach | Sustainability Dimension |
|---|---|---|---|
| [28] | Multimodal transportation networks | MCDM, DEA | Economic |
| [63] | Bus fleet network | AHP, TOPSIS | Economic, environmental |
| [68] | Submarine power cable routing selection | G-MCDA, DEA | Economic, social |
| [69] | Emergency evacuation paths of the urban metro station | TOPSIS, GRA | Social, environmental |
| [55] | Route selection in multimodal transportation networks | AHP, DEA, TOPSIS | Economic |
| [70] | Transmission network of nuclear power plant | Fuzzy AHP | Environmental, social |
| [65] [71] | Sustainable tourism paths Tourism optimal path selection | AHP, ELECTRESWOT, AHP | Economic |
| [17] | Airline new route selection | MCDM | Economic |
| [40] | Route selection in multimodal supply chains | Fuzzy MCDM | Economic |
| [53] | Path and site landfill selection | AHP, GIS | Social, economic |
| [72] | Municipal solid waste landfill siting | AHP, GIS | Environmental, socio-economic |
| [41] | Optimization of public transport networks | AHP | Economic |
| [43] | Transport service on public roads and passenger transport | AHP | Economic |
| [73] | Operational efficiencies of Turkish airports | AHP, DEA | Economic |
| [74] | Sustainable supply network optimization | DEA | Economic |
| [11] | Vehicle routing optimization model | AHP | Environmental, economic |
| [75] | Regional transport sustainability | SBM, DEA | Economic, environmental, social |

**Table 1.** *Cont.*

| Reference | Research Context | Applied Approach | Sustainability Dimension |
|---|---|---|---|
| [45] | Multimodal freight transportation systems | AHP, DEA | Economic, environmental |
| [18] | Optimal routing for mass transit systems | MCDM | Economic, environmental, social |
| [7] | Sustainable route selection of petroleum transportation | MCDM | Economic, environmental, social |
| [62] | An optimization model for sustainable transportation in the mining industry | AHP, DEA | Economic, environmental |
| [54] | Route prioritization of urban public transportation | MCDM | Economic, environmental, social |
| [76] | Public road transportation systems | AHP, DEA | Economic |
| [67] | Multicriteria route selection | AHP, TOPSIS | Economic |
| [77] | Multimodal green logistics: | AHP, DEA | Environmental, economic |
| [35] | Route selection in multimodal transportation | AHP | Environmental, economic |
| [39] | Distribution network planning reliability | AHP, DEA | Economic |
| [78] | Environmental assessment of land transportation | DEA | Environmental |
| [79] | Sustainable intermodal transport affected by COVID-19 | AHP, DEA | Environmental, economic |
| [59] | The last-mile delivery problem with service options | MCDE | Environmental, social, economic |

As reported in Table 1, and according to the literature, the integration of sustainable aspects in handling transportation route selection decision-making problems has not been widely addressed in the literature. In addition, the integration of AHP and DEA for managing sustainable transportation systems for route selection has not been found in the published research. Nonetheless, some researchers have used the current methodology for combining these two techniques in several different fields. For example, Ref. [39] used combined AHP and DEA to effectively assess if the distribution network scheduling structure meets the planning targets and needs. Additionally, Ref. [62] also used the same approach to develop a multidisciplinary integrated optimization framework for an advanced capacitated sustainable transportation for vehicle selection issues in the mining industry.

While there have been numerous studies utilizing MCDM techniques in transportation planning, some gaps remain in addressing the integrated sustainable criteria into the decision-making process. Historically, economic factors such as cost effectiveness and time savings have been prioritized above wider sustainability issues such as environmental implications, social safety, and public wellbeing. The environmental impacts of transportation, such as emissions, air pollution, and noise along with social considerations are generally overlooked and underestimated in route selection decision making. Additionally, insufficient stakeholder engagement, limited data availability, and challenges in analyzing the trade-offs between conflicting objectives have limited the integration of sustainable criteria in previous research. To address these gaps, this study presents a comprehensive approach that engages diverse stakeholders, considers a wide range of sustainability factors, and uses a real case study to create holistic and sustainable transportation route selection systems. Finally, this paper investigates waste transportation issues that, although their cruciality, have not been addressed in previous research, especially in developing countries.

## 3. Materials and Methods

### 3.1. Sustainable Waste Transportation Optimization Methodology

In the context of sustainable transportation, existing investigations have focused on environmental criteria when selecting optimal transportation routes. However, there has been a lack of consideration regarding waste transportation problems and addressing sustainable criteria. To address this gap, we propose an approach that offers comprehensive information on the most efficient route for the case of the waste transportation system. This approach helps to find a balance between environmental, social, and economic considera-

tions and government regulation by integrating different criteria. The present case study addresses the sustainable challenges of route planning in waste transportation companies in a North African country.

The waste management company is responsible for collecting household and commercial waste from various regions in urban areas. To guarantee frequent and regular waste collection, the gasoline trucks are used to follow traditional predefined routes to collect waste based on specific collection points and schedules. In fact, individuals in developing countries lack a waste sorting system. As a result, waste collection trucks are equipped to efficiently gather all types of waste, including both organic and inorganic materials, during each scheduled collection day. Domestic waste collection in the city differs from the commercial waste collection in industrial zones; therefore, separate routes and schedules are maintained for these two types of waste. Once the capacity of trucks is filled with the collected waste, they transport it to a central waste treatment facility where it undergoes sorting, recycling, and proper disposal. In many cases, the traditional waste collection routes are leading to longer travel times, increased fuel consumption, and higher air emissions [80]. This inefficiency can result from a lack of proper planning and data-driven route optimization. Additionally, inefficient waste transportation contributes to increased greenhouse gas emissions and air pollution, impacting public health and the environment [51]. In addition, waste management companies in developing countries often face budget constraints and have limited resources to invest in modern waste transportation infrastructure and technologies [5]. In this context, addressing waste transportation concerns is crucial for reducing the environmental impacts of waste disposal.

The optimization and implementation of sustainable transportation practices can reduce the carbon footprint and contribute to environmental sustainability. As a result, optimized waste transportation can lead to cost savings for waste management companies, allowing them to allocate resources to the other important areas of waste management and sustainable integration. Consequently, this study investigates the prioritization of transportation routes in the waste transportation sector using an optimization approach that combines AHP-DEA to identify the most efficient and environmentally sustainable routes for waste transportation in a medium-sized city.

Several previous studies used the integrated AHP-DEA approach to address transportation optimization concerns including road networks, supply chains, or public intermodal systems as discussed in the previous section. The combination of AHP-DEA in transportation optimization provides a powerful framework for decision makers to systematically evaluate alternatives, consider multiple criteria, and identify the most efficient transportation options [66]. This approach facilitates well-informed decision making and supports the development of effective transportation solutions by incorporating both subjective judgments and data-driven efficiency analysis, [81]. In fact, the AHP is a thorough and organized approach to address challenging decision-making issues involving several criteria and alternatives [67]. It is specifically recommended while managing preferences and subjective evaluations. In our case, route selection often involves considering various criteria like distance, time, cost, safety, and environmental considerations.

The AHP hierarchical structure allows decision makers to express their subjective preferences and judgments by assigning numerical values to different criteria [70]. Additionally, it involves pairwise comparisons of criteria and alternatives, which assist in quantifying the relative significance of various factors [12]. This process helps decision makers to clarify their preferences and priorities. On the other hand, the DEA is used to compare the effectiveness of several decision-making units [81]. It is employed to compare the efficiency of alternatives' performances. DEA is the optimal technique for evaluating and ranking distinct alternatives based on their efficiency in employing inputs to create outputs [66]. In the context of route selection, this involves analyzing routes based on multiple objectives including economic, environmental, and social aspects. It is also able to handle numerous inputs and outputs, which corresponds to the multidimensional nature of waste transportation route selection decisions that include variables such as distance, time,

cost, safety, and ecological impacts. Overall, AHP is beneficial for incorporating subjective preferences and qualitative factors in route selection decisions and DEA helps to compare efficiency-based alternatives for route selection based on several sustainable criteria.

In this paper, we have used AHP-DEA to select the most efficient transportation routes considering the sustainable performance in a waste transportation case study in a developing country. The used data in this paper entails 12 waste collection zones, where each zone has different possible collection routes that will be assessed to select the optimal ones. Consequently, considerable differences can be in waste collection in the same zone but with various collection routes, due to a variety of challenges including waste collection point sets and the scheduling of waste collection routes. A summary of the used characteristics is presented in Table 2. The used route selection criteria set considers several comprehensive aspects of a route's features including economic, environmental, and social aspects of the waste transportation sector via the incorporation of quantitative metrics such as time, distance, and cost, in addition to critical qualitative aspects including accessibility, safety, and environmental issues. These requirements guarantee an accurate evaluation that takes into account the complex nature of route selection in the context of waste collection. These criteria enable decision makers to make informed choices that balance economic efficiency, social inclusivity, environmental sustainability, and public safety, resulting in optimal route selections that meet a wide range of priorities and contribute positively to both immediate and long-term societal needs.

**Table 2.** A summary of the used data.

| | | Description | Mean | Standard Deviation |
|---|---|---|---|---|
| Economic aspects | A1: Distance (km) | The distance traveled on each route. | 64,456 | 12,602 |
| | A2: Time (h) | The time taken for waste transportation on a route. | 5.631 | 1.176 |
| | A3: Cost ($) | The cost associated with waste transportation includes fuel, employees, maintenance... | 1733.763 | 346.077 |
| | A4: Fuel Consumption (L) | Assessing the amount of fuel consumed on each route as it directly affects costs and environmental impact. | 154.326 | 32.402 |
| | A5: Energy Efficiency % | Evaluating the energy efficiency of each route, which measures how effectively energy resources are utilized during transportation. | 0.673 | 0.1424 |
| Environmental aspects | A6: Waste (tons) | The capacity of waste that each transportation route can handle. | 14.588 | 2932 |
| | A7: GHG emission(kg) | Equivalent emission of $CO_2$ and its pollution. | 96.721 | 19.661 |
| | A8: Noise Pollution (dB) | Assessing the noise pollution generated during waste transportation that impact the surrounding environment and communities. | 78.291 | 16.422 |
| Social aspects | A9: Accessibility % | The accessibility of a route ensures smooth waste transportation, avoiding roadblocks and congestion. | 0.8025 | 0.1683 |
| | A10: Safety % | Safety considerations are vital to protect both waste transportation personnel and the citizens. | 0.7284 | 0.1582 |

To identify the most efficient waste collection routes, the waste collection and treatment company employed a multidimensional approach, incorporating historical data and experience-driven findings to determine route parameters. Historical data offers valuable insights into waste generation patterns and collection point sets, while qualified expertise

provides practical information and data about local conditions and circumstances. This combined approach ensures that waste collection operations are both data-driven and responsive to real-world conditions, resulting in cost-effective and efficient services adapted to the community's needs.

The first step is to identify the relevant crucial criteria for evaluating waste transportation routes. In this regard, a team of experts and stakeholders from the waste management company is involved in performing pairwise comparisons of the criteria using the AHP technique. Table 3 presents the members' details of the team responsible for sustainable transportation projects. The experts provide their subjective judgments on the relative importance of each criterion compared to the others. Thus, AHP synthesizes these judgments to obtain the relative weights of the criteria, leading to the aggregated AHP weighted score for each transportation route, reflecting its priority in the system. Therefore, the DEA is applied to assess the efficiency of each transportation route on the inputs and outputs, yielding the DEA efficiency score. We used the DEAP computer program to calculate the efficiency scores. These scores are integrated to assist in identifying the most efficient and sustainable routes, assisting decision makers in making informed choices for waste transportation planning. Figure 1 summarizes the different steps of the adopted approach.

**Table 3.** Experts and specialist details.

| Members Function | Number of Experts | Years of Experience |
|---|---|---|
| Experts from waste treatment and transportation company | 4 | 10–20 |
| Industrial transportation specialist | 5 | 10–15 |
| Transportation engineers | 4 | 5–10 |
| Environmental specialist | 3 | 8–10 |
| Data analytics operators | 5 | 5–10 |
| Stockholders and government representatives | 3 | 10–15 |

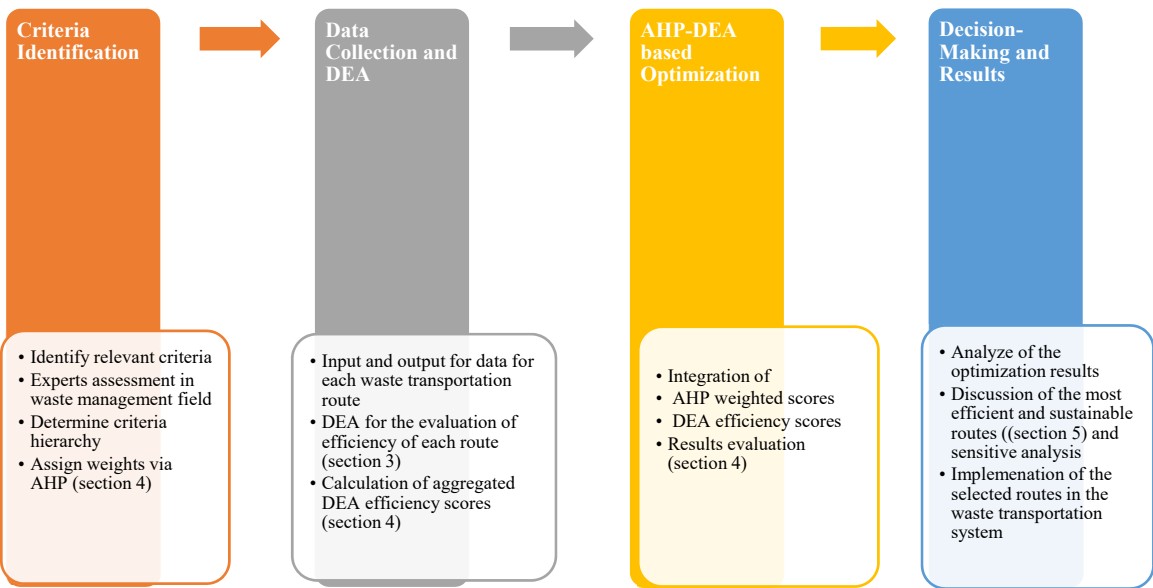

**Figure 1.** The adopted methodology flowchart.

## 3.2. AHP Evaluation for Route Selection

The AHP decision-making method is a prevalent MCDM approach for resolving challenging decision-making issues [66]. The key components of the AHP approach involve

dividing decision making into manageable related issues at each level of the decision hierarchy, assigning the features at each degree of the structure of decisions and combining targets to establish the entire importance of the alternatives to decision making [70].

Evaluating and prioritizing transportation routes is challenging as sustainability factors are integrated into a conventional economics paradigm. In this context, assessing sustainability features is an essential challenge when reconciling economic and environmental goals [79]. Initiatives have already attempted to establish parameters for the selection of route planning issues; nevertheless, existing comprehensive frameworks are limited in terms of sustainable criteria. The AHP approach may be employed to select alternate transportation routes to effectively use energy assets and prompt less environmental damage. For example, Refs. [65,67] developed a set of criteria for selecting a suitable one. Additionally, Ref. [62] employed an AHP-based model to determine the best mode of sustainable transportation for logistical operations.

In this paper, the AHP approach is used to calculate the important weights of routes. From a sustainability standpoint, we analyzed several important aspects such as sustainable criteria, transportation features, route characteristics, and vehicle features in waste transportation systems. Furthermore, we examined the opinions of specialists, including experts from the waste management sector, specialists from the transportation industry, and stockholders' representatives as presented in Table 3 concerning the experts and specialist details. In addition, experts perform pairwise comparisons between the chosen criteria to establish their relative importance using a defined scale from 1 to 9 (highly to less important) to quantify the relative importance of one criterion over another.

We created a full set of criteria for the sustainable evaluation of routes based on extensive literature reviews. The initially developed criteria selection was presented to the organization's professional decision-making committee, and therefore, responses and proposed improvements were received. The final set of requirements included universally acknowledged criteria. The primary categories of the selected criteria are economic including energy efficiency, fuel consumption, time, distance, and cost; environmental, including waste recycling, noise pollution, and GHG emission; and social, including safety and accessibility.

The AHP process synthesizes the pairwise comparison judgments to calculate the relative weights of the criteria [71]. The weights represent the importance of each criterion concerning the objective for waste transportation route selection. Therefore, the aggregated AHP weighted score is integrated to calculate the DEA efficiency to identify the optimal waste transportation routes that satisfy the given constraints while achieving the highest level of efficiency and sustainability.

*3.3. DEA Parameters Analysis*

Data Envelopment Analysis is an analytical method developed in 1978 as a non-parametric approach to evaluate the relative efficiencies of a set of similar Decision-Making Units (DMUs) [82]. These DMUs could represent organizations, institutions, or companies, and DEA aims to assess their performance based on multiple inputs and outputs [36]. The technique is widely recognized and utilized for conducting efficiency comparisons across various domains, including but not limited to the transportation sector [66]. In addition to its widespread application, DEA has also been extended and enhanced by researchers in various fields. The DEA approach enables a more comprehensive assessment of transportation efficiency, providing valuable insights for optimizing transportation systems and decision-making processes [75].

In our research, we employ DEA as a method to conduct a comparative efficiency analysis of various waste transportation routes. This analysis is essential given the significant variations in operational, environmental, and economic strengths and weaknesses among different transportation routes, particularly considering the different regions where waste bins are often situated. The location of these regions leads to diverse efficiency scores not only for different distances, durations, and times, but also for the constraints of different

economic, social, and environmental issues. Thus, to achieve a comprehensive assessment of route efficiency, the DEA method is used. This approach enables the identification of the most efficient route for waste transportation by maximizing the efficiency scores of transportation routes and considering diverse criteria. Consequently, it helps in making informed decisions for optimizing transportation systems while optimizing environmental, economic, and social impacts.

In this case, each transportation route will be treated as a DMU to calculate its efficiency score to determine the efficiency of each route in the waste transportation system. The objective is to select routes that achieve the highest level of efficiency while satisfying sustainable waste management constraints. The proposed approach aims to find the optimal routes that maximize efficiency while optimizing social, economic, and environmental features. In this context, for each transportation route, we will calculate its efficiency score using DEA. The efficiency score measures how efficiently the transportation route utilizes its inputs to produce outputs relative to the other routes in the system. The DEA model will have input factors and output factors for each transportation route. Routes with higher efficiency scores are considered more efficient and can be given preference in the waste transportation system. The following are definitions of the parameters and decision variables employed in this approach:

Route-i: i-th number of transportation routes (index: $i = 1, 2..., m$);

Input: j-th number of input factors (e.g., truck waste capacity, distance, time, safety, accessibility, and fuel consumption) (index: $j = 1, 2..., n$);

Output: k_th number of output factors (e.g., cost, GHG emission, noise pollution, and energy efficiency) (index: $k = 1, 2..., s$);

$X$: m $\times$ n matrix representing the input data for each route.

($X_{ij}$ is the value of the $j$-th input, for the $i$-th route).

$Y$: m $\times$ s matrix representing the output data for each route.

($Y_{iq}$ is the value of the $q$-th output, for the $i$-th route).

$$\text{Efficiency score } \theta = \frac{\text{weighted sum of outputs}}{\text{weighted sum of inputs}}$$

Maximize the efficiency score $\theta$ of a route:

$$\text{Max } \theta \text{ subject to } \sum \alpha_q * Y_{iq} \leq \sum \beta_j * X_{ij} \quad (1.1)$$
$$\text{for all } j \text{ and } q \text{ (input constraint)}$$

$\theta$ is the efficiency score representing how well the route uses its inputs to produce outputs relative to the other routes.

$\alpha_q$ is the weight given to output q, and $\beta_j$ is the weight given to input j. These weights are non-negative and indicate the importance of each route in the efficiency calculation.

Constraint 1 ensures that the weighted sum of inputs for each route is less than or equal to the weighted sum of inputs for any efficient ($\theta = 1$) route.

Constraint 2 ensures that the weighted sum of outputs for each route is greater than or equal to the weighted sum of outputs for any efficient ($\theta = 1$) route.

The DEA model allows us to obtain the efficiency score ($\theta$) and the weights for each route. The efficiency score ($\theta$) of each route ranges from 0 to 1. A score of 1 indicates full efficiency, while scores less than 1 suggest varying degrees of inefficiency. Efficient routes represent the best choices for the given set of criteria, and inefficient routes may require improvements or adjustments to become more efficient. In conclusion, DEA is a mathematical technique that assesses the efficiency of multiple entities by comparing their inputs and outputs concerning the most efficient entities. The approach seeks to maximize efficiency while taking into account the limitations imposed by the efficient entities, while also maintaining a balance between the inputs and outputs.

## 4. Results

In this section, we present the results of the proposed approach in three phases. First, we present the AHP evaluation of routes. Second, we present the DEA evaluation of the comparative efficiencies of transportation routes. Finally, we discuss the results of waste transportation systems' route selection.

After defining the criteria used in this research to evaluate waste transportation routes, we performed a pairwise comparison for each criterion. We provided a relative weight representing the importance of one criterion compared to the others. The scale for pairwise comparisons in the waste transportation sector has been defined from 1 to 9, where 1 means both criteria are equally important, and 9 means one criterion is extremely more important than the other. This has resulted in pairwise comparison matrices for all the criteria, which allow the calculation of the priority weights for each criterion using the AHP method. This involves calculating the geometric mean of each row in the matrices to obtain the priority vector. Then, we normalize the priority vectors to sum up to 1 for each criterion. Additionally, we performed a consistency check to ensure that the pairwise comparisons are reasonable and consistent. The consistency ratio (CR) is calculated to check for consistency [83]. In this regard, our results of the CR value were all less than 0.1, which is considered acceptable. Table 4 presents the results of the weight score of the different criteria.

**Table 4.** Weight scores of the used criteria.

| Criteria | (A1) | (A2) | (A3) | (A4) | (A5) | (A6) | (A7) | (A8) | (A9) | (A10) |
|----------|------|------|------|------|------|------|------|------|------|-------|
| Weight | 0.123 | 0.104 | 0.142 | 0.131 | 0.151 | 0.122 | 0.08 | 0.07 | 0.06 | 0.03 |

The results of the AHP analysis reveal the relative importance of each criterion in the waste transportation route prioritization. The criterion with the highest weight is (A5), which indicates that optimizing energy utilization during waste transportation is considered the most crucial factor in the decision-making process. This could indicate selecting routes with efficient energy consumption and low energy during transportation. The second most important criterion is (A3) and (A4), indicating that cost-effective and fuel-efficient routes are desired to minimize operational expenses. Next, (A1) and (A2) are prioritized, which implies that shorter route distances and times are preferred to reduce transportation time and costs. In addition, (A6) and (A7) are also considered important factors, suggesting that safer and easily accessible routes with lower greenhouse gas emissions and maximum waste capacity are given significant consideration.

The following step entails the DEA evaluation of comparative efficiencies of transportation routes. In fact, in the DEA analysis, the efficiency score is a measure that assesses the relative efficiency of DMUs in the used dataset. It helps to determine how efficiently each DMU utilizes its inputs to produce outputs, compared to the other DMUs in the dataset [35]. The selection of inputs and outputs are essential components of the DEA model and they define the performance evaluation of the adopted framework [60].

In this case, we consider attributes that contribute to the waste transportation process as inputs, including distance, time, amount of waste, safety, accessibility, and fuel consumption. For the outputs, we defined the following criteria (cost, GHG emission, noise pollution, and energy efficiency). The efficiency scores of various transportation routes in every zone have been calculated via the use of the output maximization DEA models outlined in Section 3.3, as shown in Table 5.

**Table 5.** AHP-DEA approach results.

| | Routes Weights | | Routes Weights |
|---|---|---|---|
| Zone A: Ait Oulal | R1: 0.483<br>R2: 0.678<br>R3: 1.000<br>R4: 0.478<br>R5: 0.632<br>R6: 0.305<br>R7: 0.518 | Zone F: Agdal | R1: 0.768<br>R2: 0.235<br>R3: 0.738<br>R4: 0.621<br>R5: 0.605<br>R6: 1.000 |
| Zone B: Menzeh | R1: 1.000<br>R2: 0.550<br>R3: 0.766<br>R4: 0.655<br>R5: 0.597<br>R6: 0.498 | Zone I: Toulal | R1: 0.753<br>R2: 0.625<br>R3: 0.588<br>R4: 0.606<br>R5: 0.602<br>R6: 1.000 |
| Zone C: Hamria | R1: 0.621<br>R2: 0.572<br>R3: 0.602<br>R4: 1.000<br>R5: 0.537<br>R6: 0.654<br>R7: 0.629 | Zone J: Al Mansour El Jadid | R1: 0.744<br>R2: 0.672<br>R3: 1.000<br>R4: 0.678<br>R5: 0.649<br>R6: 0.619<br>R7: 0.539 |
| Zone D: Hay El Houda | R1: 0.689<br>R2: 1.000<br>R3: 0.633<br>R4: 0.592<br>R5: 0.517<br>R6: 0.643 | Zone K: Zitoun | R1: 0.713<br>R2: 1.000<br>R3: 0.631<br>R4: 0.715<br>R5: 0.572<br>R6: 0.689<br>R7: 0.521 |
| Zone E: Al Massira | R1: 0.612<br>R2: 0.675<br>R3: 0.503<br>R4: 0.723<br>R5: 1.000<br>R6: 0.729 | Zone L: Al-Amal | R1: 1.000<br>R2: 0.699<br>R3: 0.508<br>R4: 0.647<br>R5: 0.321<br>R6: 0.534<br>R7: 0.651 |
| Zone N: Ain Ourma | R1: 0.577<br>R2: 1.000<br>R3: 0.753<br>R4: 0.512<br>R5: 0.675<br>R6: 0.561 | Zone M: Medina | R1: 0.656<br>R2: 0.644<br>R3: 0.715<br>R4: 1.000<br>R5: 0.688<br>R6: 0.533<br>R7: 0.638 |

In Table 5, we present the DEA-AHP approach results, which combine the AHP weights with the DEA efficiency scores. The output of the results presents the most efficient route among all routes in every zone, of which these efficiency scores are recalculated considering the weights from the AHP analysis. The DEA-AHP approach helps to integrate the relative importance of the criteria (AHP) with the efficiency evaluation (DEA) to obtain a more comprehensive and informed decision. For example, in zone A, route 3 has the highest efficiency score of 1.000, which is the most efficient and optimal route among the given options.

A closer look at the results of the first zone, for example, reveals that the DMUs have different efficiency scores. For example, the lower score of R1 and R6 indicates that these routes are not utilizing their inputs efficiently to achieve the desired outputs. It means that the route is relatively inefficient in managing costs, controlling GHG emissions, and optimizing energy consumption regarding the consumed time, distance, and amount of collected waste. However, Route R2 has a higher efficiency score compared to R1, which suggests that it performs better in managing its inputs to generate the desired outputs. It is more cost-effective, emits fewer GHGs, and uses energy more efficiently. However, R3 has achieved optimal efficiency in utilizing its inputs to produce optimal outputs. It is cost-effective, has minimal GHG emissions, and optimizes energy consumption during the waste transportation process.

Similar to R1, R4, R6, and R7 which also have low-efficiency scores, it indicates inefficiency in using inputs to produce outputs. There are opportunities for improvement in cost management, GHG emissions reduction, and energy optimization for this route. At

the same time, the routes R5 and R7 have a moderate efficiency score, which means it is relatively more efficient compared to R1 and R4 but still has scope for improvement. It is cost-effective, emits fewer GHGs, and uses energy efficiently, but not to the extent of achieving perfect efficiency.

Overall, the efficiency scores show that there are varying levels of resource efficiency among the waste routes. Routes R3 and R2 are the most efficient, while R6 is the least efficient. These efficiency scores provide valuable insights into route performance and serve as a basis for decision making to optimize waste route selection, resource allocation, and waste management strategies for improved cost effectiveness, environmental sustainability, and energy efficiency. Indeed, routes with lower efficiency scores may require adjustments and enhancements to align with sustainable goals and maximize resource utilization.

In this research, a sensitivity analysis was conducted to validate the accuracy of the route ranking. It assesses the impact of small changes in input variables within a given point in the range of parameters [83]. Sensitivity analysis for route selection within the context of MCDA is a significant tool for analyzing the flexibility and reliability of the selected route decisions in different scenarios. This method enables decision makers to evaluate the impact of variations in criterion weights and information regarding final route decisions. As a result, executing sensitivity analysis on the outcomes of a decision issue might offer beneficial details to the decision maker, thereby allowing them to make more informed choices. The goal of sensitivity analysis in the present research is to increase and decrease the value of every criterion while decreasing/increasing the weighting of the remaining criteria similarly; therefore,10 scenarios for 10 criteria are investigated using simulation with a balanced weighting of criteria. For this intention, adjustments have been made to the extremely high or low values of each criterion, resulting in 10 distinct experimental combinations employed on the route alternatives. As a consequence of the ten scenarios, the sensitivity analysis findings for Zone A demonstrate that, regardless of the weight of the criterion, R3 emerged as the optimal waste transportation route choice for all experiments (Table 6). In tests 2, 4, and 8 concerning alternatives R1, R2, and R4, R6 experienced slight alterations in their priority order. This demonstrates that in these tests, the criteria importance of those options is sensitive. Overall, although the significance of the measurement of weight varies slightly, the order of preference remains particularly the same. The findings validate the suggested assessment framework's reliability. The sensitivity analysis findings are provided in Table 6 for Zone A, and the illustration of these results is provided in Figure 2. Appendix A presents the sensitive analysis of the other remaining zones.

**Table 6.** Ranking of sensitivity analysis for routes in Zone A.

| Scenarios | R1 | R2 | R3 | R4 | R5 | R6 | R7 |
|:---:|:---:|:---:|:---:|:---:|:---:|:---:|:---:|
| 1 | 5 | 2 | 1 | 6 | 3 | 7 | 4 |
| 2 | 6 | 2 | 1 | 7 | 4 | 5 | 3 |
| 3 | 5 | 3 | 1 | 6 | 4 | 7 | 2 |
| 4 | 5 | 3 | 1 | 7 | 4 | 6 | 2 |
| 5 | 5 | 2 | 1 | 6 | 3 | 7 | 4 |
| 6 | 6 | 2 | 1 | 5 | 3 | 7 | 4 |
| 7 | 5 | 2 | 1 | 6 | 3 | 7 | 4 |
| 8 | 7 | 3 | 1 | 5 | 4 | 6 | 2 |
| 9 | 5 | 2 | 1 | 6 | 3 | 7 | 4 |
| 10 | 5 | 2 | 1 | 6 | 3 | 7 | 4 |

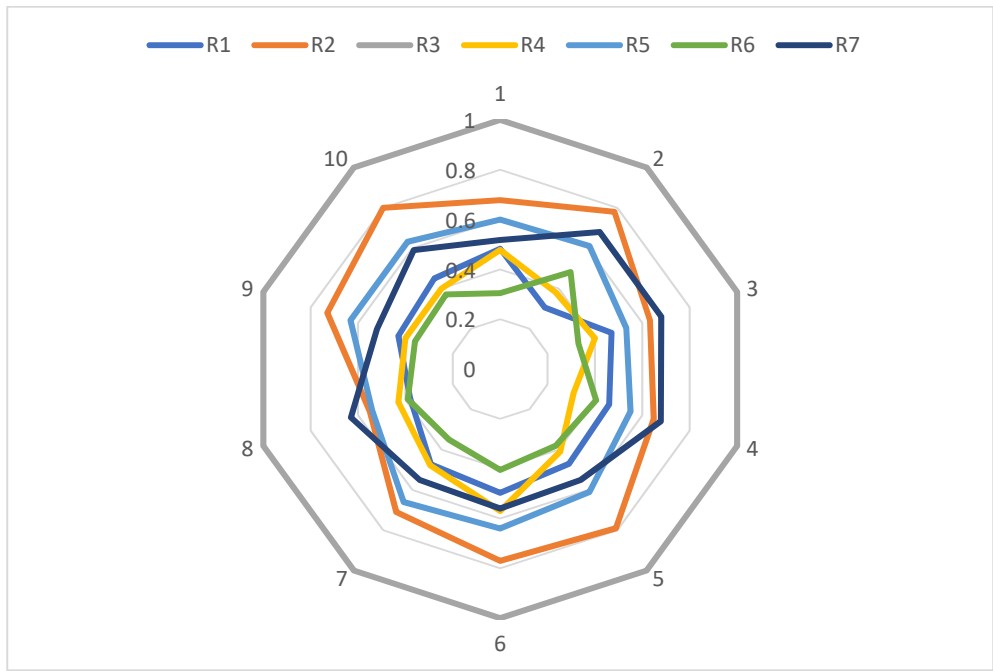

**Figure 2.** Sensitivity analysis for Zone A.

This AHP-DEA approach provides a comprehensive and informed decision-making process for waste transportation route selection in the main area of waste collection zones in a medium-sized city. The AHP analysis allows decision makers to consider the relative importance of criteria based on experts' judgments, reflecting their preferences and expertise in the waste management sector. The DEA analysis then evaluates the efficiency of each route in the given zone of waste transportation, enabling an objective comparison of their performances.

## 5. Discussion

The findings indicate a list of effective waste transport routes for different regions. This analysis enables executives to evaluate the different possible transportation paths for waste collection from various environmental and economic perspectives.

The results of this research, which combines the AHP and DEA approach to prioritize transportation routes based on multiple criteria, including distance, time, safety, cost, and environmental features, present several potential discussions and implications. In fact, the research demonstrates the importance of considering sustainable criteria when selecting transportation routes for waste management. The study highlights the potential for waste management companies to make environmentally responsible choices while optimizing their transportation operations by incorporating environmental factors, such as noise pollution, GHG emissions, and energy efficiency. Indeed, by selecting efficient routes, the company can lower operational costs while contributing to environmental conservation efforts, which can lead to reduced fuel consumption and emissions. These findings highlight the need to achieve a balance between economic, social, and environmental factors in waste transportation systems. This research offers decision makers a thorough framework to assess alternative solutions that minimize costs while simultaneously having a lesser environmental impact, leading to creating a more sustainable waste management strategy.

Developing countries often face limited waste management infrastructure and resources. The proposed research could provide insight into how to overcome these limitations and demonstrate the feasibility and effectiveness of adopting sustainable concepts in waste transportation systems. Moreover, while developing countries might perceive sustainable waste transportation practices as expensive, this research can demonstrate that the long-term cost savings from improved efficiency and reduced environmental damages

outweigh the initial investments of sustainable integration. Also, AHP-DEA analysis can be highly advantageous for waste management companies in developing countries facing budget constraints and limited resources. In addition, developing countries often face environmental challenges due to rapid urbanization and industrialization [84]. Thus, incorporating sustainable waste transportation practices aligns with organizations' commitment to sustainable development and environmental responsibility. Therefore, this research can help policymakers and waste management stakeholders to understand the positive impact of adopting sustainable transportation strategies. At the same time, the findings of the research could inform waste management policies and regulations to design and impose standards that encourage waste management companies to adopt more sustainable transportation practices. Additionally, taking into account the possible route selection in waste transportation services in various regions enables an analysis of the geographic distribution of routes. This analysis can help to determine whether certain regions are more significantly impacted by the negative effects of waste transportation such as pollution and health risks.

After comparing the most efficient route for each zone, the findings indicated that distance is a prevalent characteristic among efficient routes. Indeed, distance is a critical factor in determining the efficiency of road routes, including waste collection, due to its direct impact on cost savings, time efficiency, resource optimization, environmental impact, and overall operational efficiency. In this context, by minimizing the waste collection distance, the waste collection operations can lower operational expenses, provide faster and more reliable services, reduce their carbon footprint, maintain consistent schedules, and make optimal use of the collection vehicles, thus benefiting both service providers and the communities.

The results of this research are found to be highly compatible with other studies from the existing literature in the same context. Indeed, multiple independent investigations on transportation and sustainability assessment consistently identified and emphasized the significance of sustainability criteria in transportation decisions [85]. For instance, [80] highlighted the importance of considering sustainability factors in transportation planning to achieve long-term environmental benefits and minimize negative impacts on communities. Moreover, [78] emphasized the need to promote low-carbon and sustainable transportation options to reduce greenhouse gas emissions and combat climate change. Furthermore, [54] highlighted the benefits of investing in energy-efficient and low-emission public transportation options for sustainable urban road transport. Indeed, these highlighted studies have emphasized the significance of boosting low-carbon and energy-efficient transportation as an approach to achieving sustainable goals. In this context, our study shares a core commitment to improving sustainability and lowering environmental impacts via optimal and sustainable waste collection routes.

Overall, aligning transportation planning challenges, economic considerations, and environmental impact assessments across various studies further strengthens the importance of this study, particularly in developing countries. The convergence of policy implications and demonstrated economic and environmental benefits of adopting sustainable waste transportation practices highlights the methodology's relevance and effectiveness. Ultimately, the consistency with previous research from the literature demonstrates the robustness of the combined AHP-DEA approach in waste transportation route prioritization and selection and emphasizes the significance of integrating environmental and economic considerations in waste management strategies. As a result, the multi-criteria approach operates as a guide for decision makers in determining the most efficient and sustainable path to waste transport management while simultaneously considering operational and economic perspectives.

The current situation of technological innovations in waste transportation and management sectors is recognized as unsustainable in developing countries due to inadequate infrastructure, limited technological adoption, insufficient regulations, and low awareness and reliance on traditional waste management strategies [23,48,86]. At the same time,

resource constraints, rapid urbanization, and population growth further exacerbate the challenges. Consequently, to achieve sustainability, these countries must elaborate innovative approaches, invest in relevant technologies, enforce regulations, raise public awareness, and seek cooperation and support from stakeholders and the government [14,50]. Therefore, the transportation system's economic, environmental, and social impact presents a considerable challenge for experts, policymakers, and managers. They must effectively balance the competitive waste transportation demands while striving to maintain a sustainable transportation system that supports environmental, economic, and ecological wellbeing. This study established a proposed approach for transportation prioritization that highlighted the components affecting sustainability in the field of waste transportation. Waste management and treatment companies are often situated in isolated areas, while the collection and transportation of waste are conducted across cities and among citizens. This highlights the importance of a holistic and integrated approach to addressing waste transportation challenges in developing countries, rather than relying on traditional and convolutional solutions. Such an integrated approach is vital to ensure efficient, cost-effective, and environmentally sustainable waste collection in these regions [86,87]. Consequently, adopting the proposed approach in this research enables decision makers to develop an organized method for assessing and classifying waste transportation routes, which is based on relevant sustainable criteria in the context of the given case study. Transportation routes' prioritization assists in minimizing emissions and noise pollution, promotes social safety, decreases associated costs, and increases energy efficiency.

*5.1. Theoretical and Managerial Implication*

The theoretical implications of the AHP-DEA approach for the selection of sustainable transportation routes for waste management are significant in advancing our understanding of decision-making processes in the complex and multidimensional context of waste transportation systems in emerging countries. This approach contributes to the development of sophisticated decision support systems, by integrating DEA's efficiency analysis with AHP's ability to incorporate stakeholder preferences and criteria weighting. Theoretical research on this integrated approach enriches the literature on sustainable waste management and transportation planning, offering insights into how to achieve a balance between economic, environmental, and social factors. It also emphasizes the significance and importance of data-driven and evidence-based methodologies in addressing sustainability challenges in waste transportation.

From a managerial perspective, the implications of the AHP-DEA approach are revolutionary for waste management companies, policymakers, and urban planners in developing countries. The evaluation of waste transportation routes allows managers to identify and adopt the most efficient and sustainable practices, leading to cost optimization, reduced environmental impacts, and improved resource utilization. Managers can align waste transportation strategies with the preferences of various stakeholders, resulting in increased community engagement and social acceptance. Additionally, the approach enables decision makers to develop regulations that incentivize sustainable transportation practices, promoting an ecological and more resilient waste management system. Subsequently, adopting the DEA-AHP approach can enhance operational efficiency, support regulatory compliance, and boost the environmental performance of waste transportation practices, all while aligning with broader sustainability goals.

Overall, integrating sustainability into decision making has a significant impact on decisions by increasing the criteria beyond economic factors to include environmental and social considerations, promoting a long-term perspective that anticipates future consequences, enabling proactive risk mitigation, promoting stakeholder engagement, and enhancing reputation and sustainability compliance with evolving environmental regulations. This comprehensive decision-making approach not only benefits organizations by reducing negative consequences and maximizing resource usage, but it also aligns with society's values and solves major environmental and social concerns.

The managerial implications are equally substantial since the approach enables waste management stakeholders to make informed and evidence-based decisions, driving more sustainable and environmentally responsible waste transportation practices. This approach serves as a bridge between theory and practice, providing a significant approach to improving waste management strategies and contributing to a more sustainable future.

### 5.2. Limitations and Future Works

While the approach of using AHP-DEA for selecting sustainable transportation routes for waste management offers valuable insights, it is essential to acknowledge its limitations. These limitations could be related to technological limitations, regulatory barriers, and financial constraints, especially in developing countries. First, the AHP involves subjective judgments and pairwise comparisons by decision makers to determine the relative importance of the criteria. The investigation of the criteria in this research focuses on economic and environmental criteria more than social aspects. Second, the DEA analysis requires a sufficient number of DMUs to create a reliable efficiency frontier [63]. Generally, in waste management contexts, there is a limited number of transportation routes for waste management companies, which could impact the robustness of efficiency scores. Third, the findings are specific to the medium-sized city and waste management company. As a result, addressing the trade-offs between the economic and environmental criteria can be challenging with larger cities' size and different weighting methods might lead to alternative results. Thus, proceeding with the results to the other fields with different waste management contexts requires adaptation. Furthermore, the efficiency and sustainability of waste transportation routes might be influenced by external factors that have not been addressed in this study, such as traffic congestion, weather events, or changes in waste disposal regulations.

Despite these limitations, the study provides valuable insights into the potential benefits of integrating sustainability criteria into waste transportation route selection. Acknowledging these constraints can enable future research to use this research, to further refine and enhance the approach and explore new avenues, making it more applicable and beneficial for waste management decision makers. For example, conducting comparative case studies across different cities or regions with varying waste management contexts would assess the transferability and generalizability of the DEA-AHP approach and identify factors that influence its performance in different settings. Moreover, incorporating a life cycle assessment to assess the overall environmental impact of waste transportation routes considering the entire life cycle of waste management from collection to disposal would also provide a more holistic view of the routes' environmental implications. In addition, future research can explore the use of real-time data and advanced technologies to monitor waste transportation routes continuously and optimize them based on changing conditions and disruptions. This could lead to adaptive and dynamic route planning, further enhancing efficiency and sustainability. Further studies can investigate the potential of emerging technologies, such as electric vehicles, enabling tracking technologies (e.g., Internet of Things, Big Data Analytics) in improving the sustainability of waste transportation routes. Moreover, integrating the methodology with circular economy principles may present further opportunities for optimizing sustainable waste transportation routes.

### 6. Conclusions

This research contributes to the literature on sustainable transportation systems' development by presenting the effectiveness of the integrated DEA-AHP approach for selecting sustainable transportation routes in the waste transportation sector. This study presents a decision-making framework that combines the strengths of DEA's efficiency analysis and AHP's criterion weighting, offering a comprehensive approach to enhancing waste transportation operations. This research introduces an innovative method for prioritizing efficient transportation routes across different regions. In this regard, the study has been conducted within the context of a developing country, specifically in a medium-sized

city, where conventional waste management methods are still prevalent. The presented approach achieves the dual goals of optimizing resource utilization and minimizing environmental impacts, resulting in reduced emissions and cost savings. The outcomes of this methodology present significant implications, where it advances the understanding of sustainable waste management decision making and supports waste management companies and managers. These evidence-based tools can promote environmental, social, and economical efficient waste transportation practices.

**Author Contributions:** Conceptualization, H.H., A.B., A.C. and N.H.; Methodology, H.H., A.B., A.C. and N.H.; Validation, H.H., A.B., A.C. and N.H.; Formal analysis, H.H., A.C. and N.H.; Writing—original draft, H.H.; Writing—review and editing, H.H., A.B., A.C. and N.H.. All authors have read and agreed to the published version of the manuscript.

**Funding:** This research received no external funding.

**Informed Consent Statement:** Not applicable.

**Data Availability Statement:** The data presented in this study are available on request from the corresponding author. The data are not publicly available due to confidential issues.

**Conflicts of Interest:** The authors declare no conflict of interest.

## Appendix A. The Sensitive Analysis of the Studied Zones

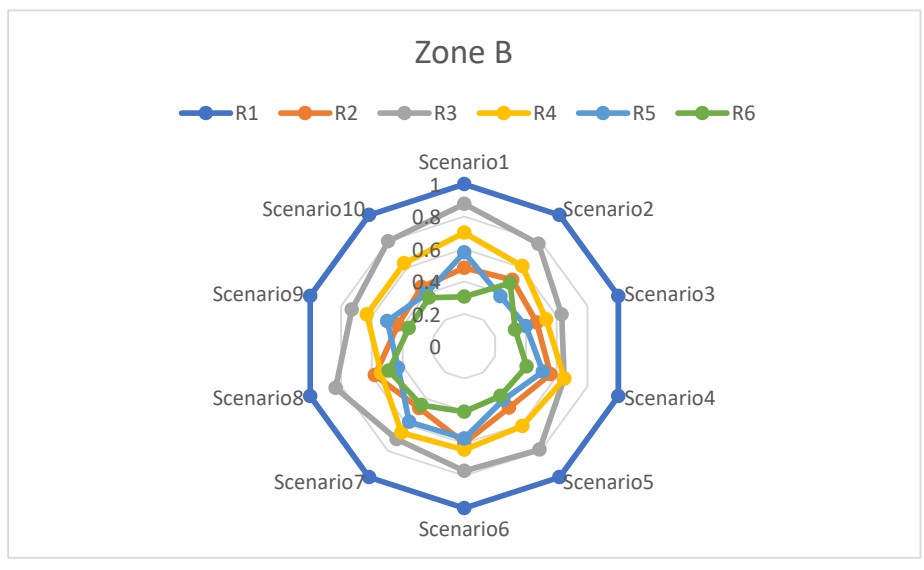

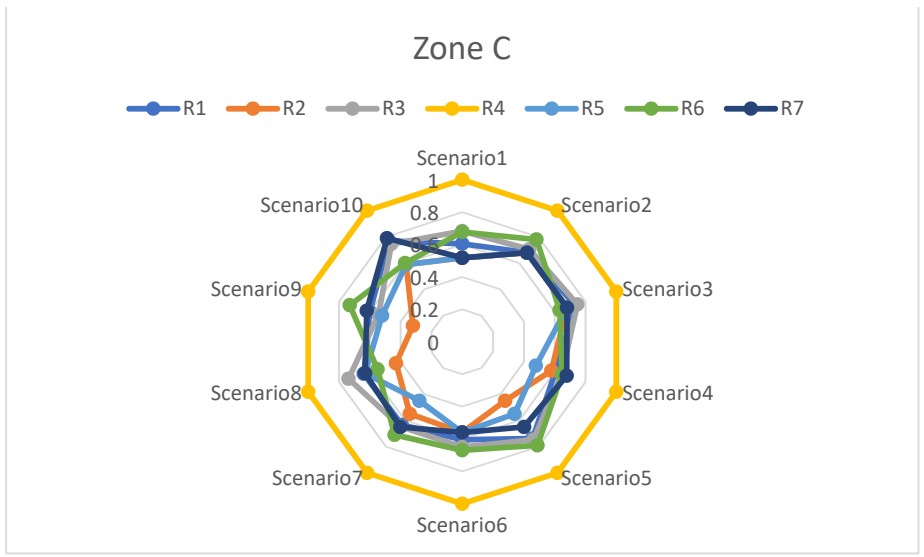

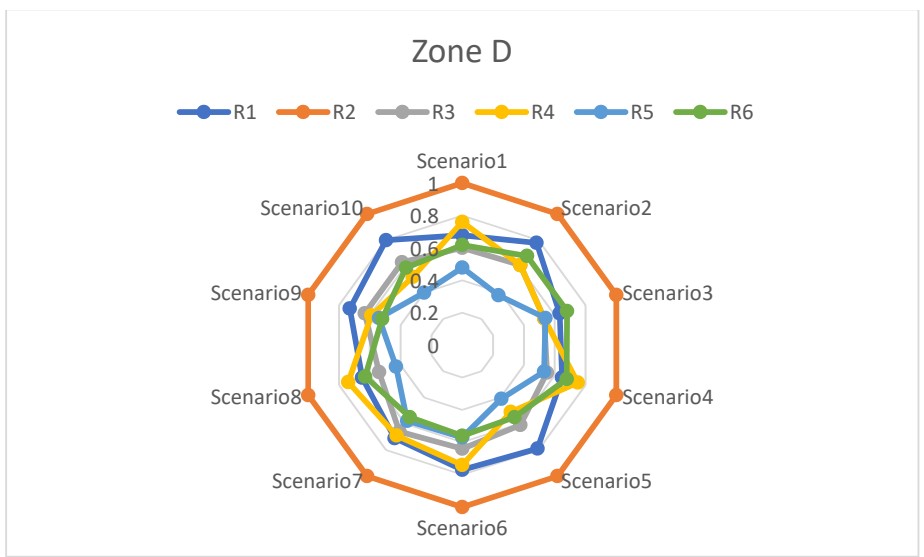

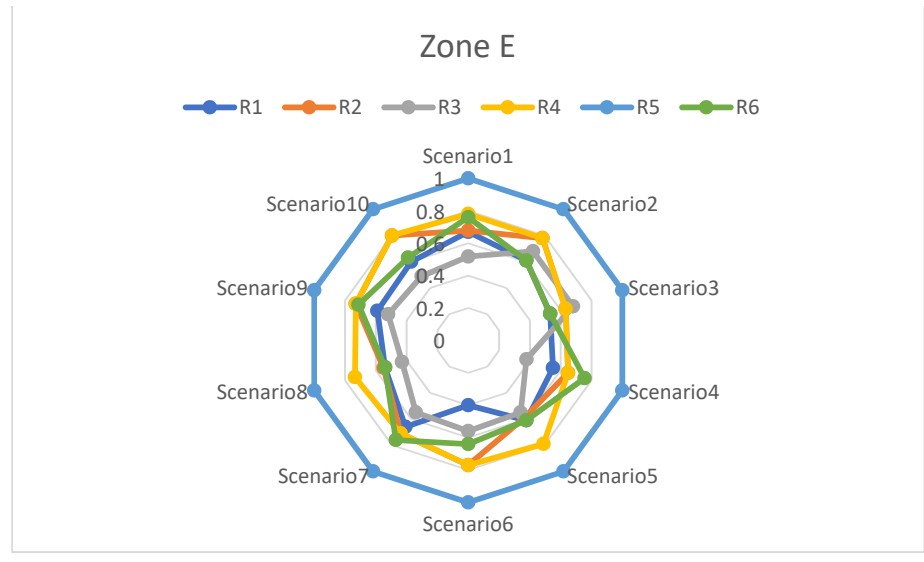

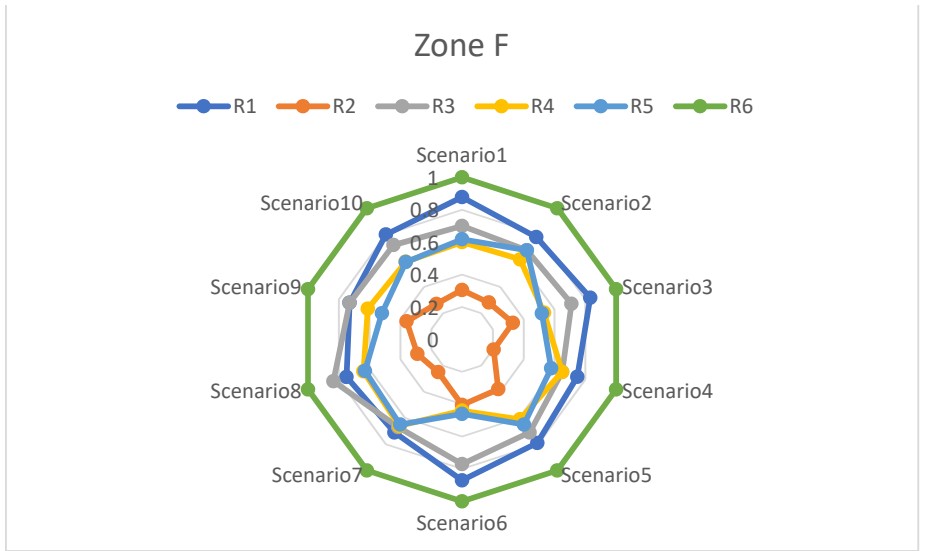

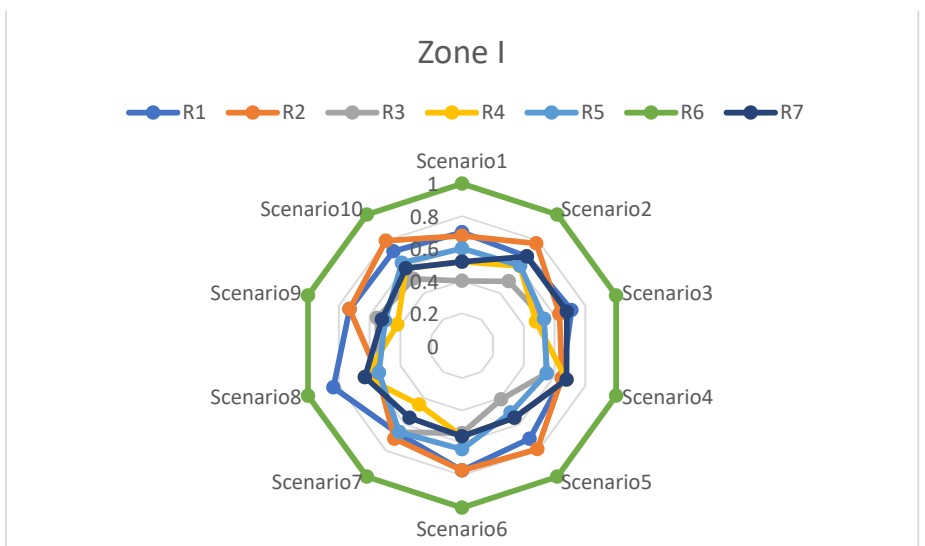

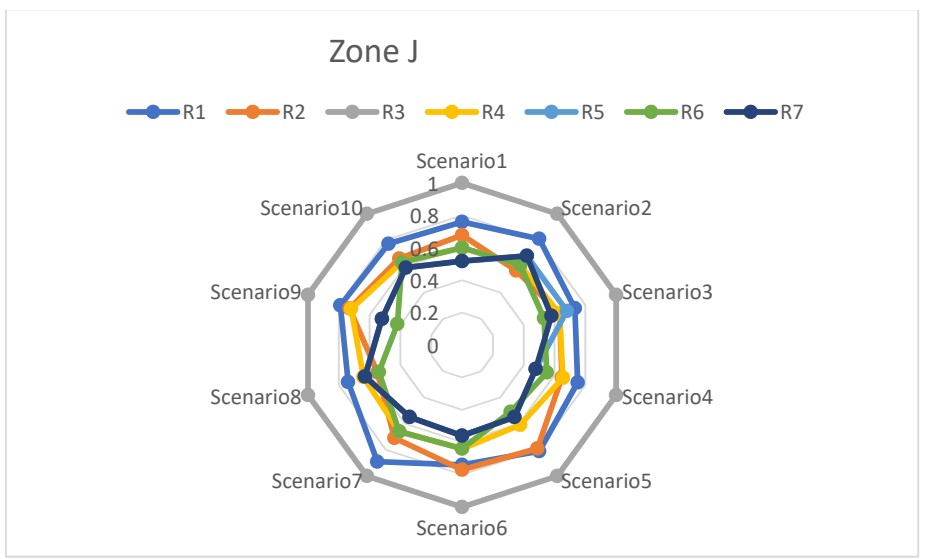

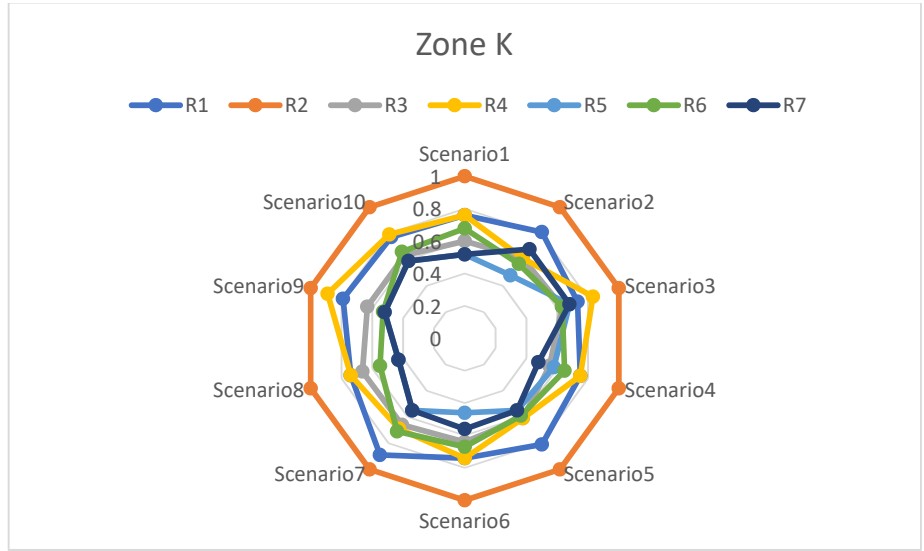

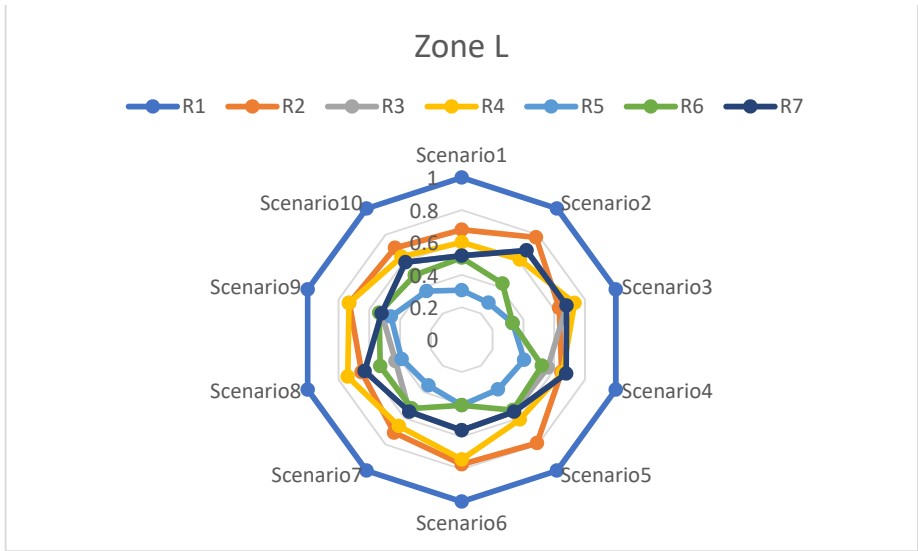

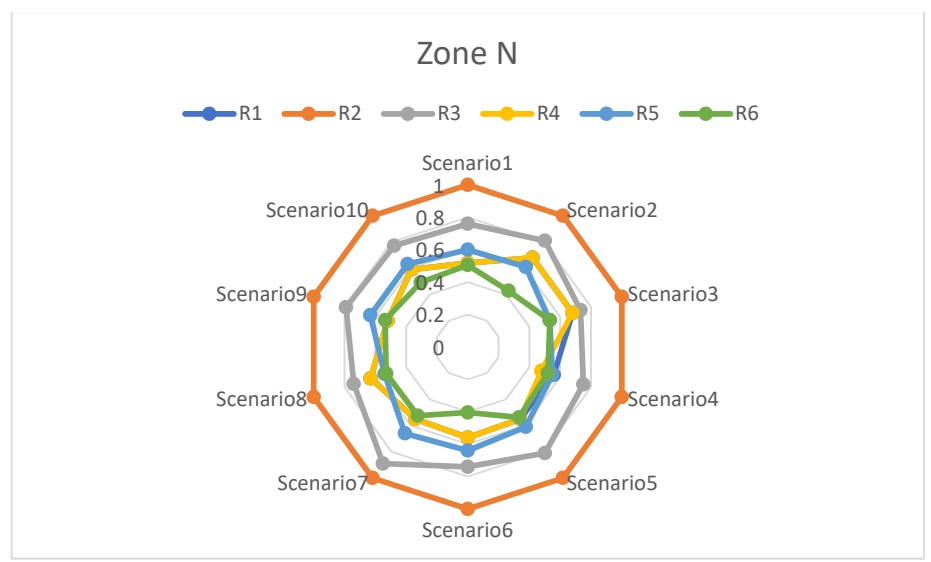

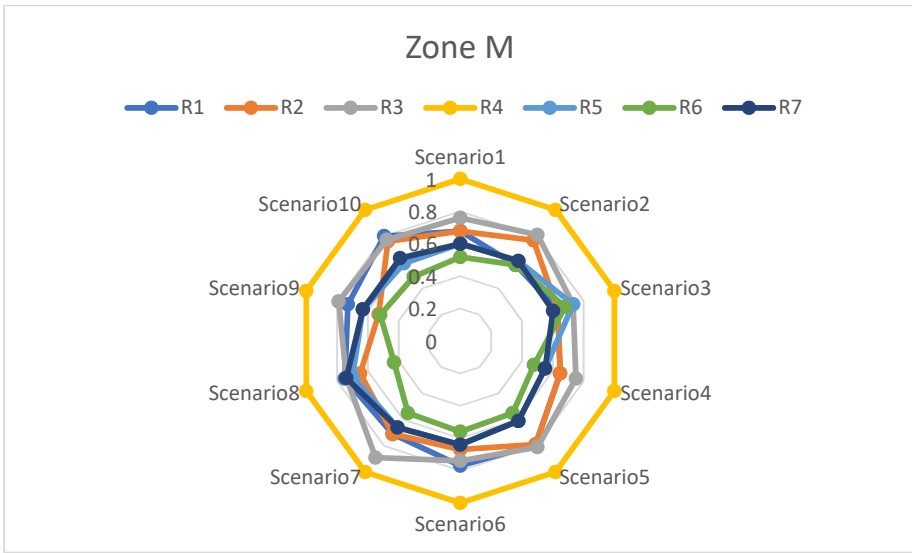

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
