# Peer review of "Achieving a Sustainable Transportation System via Economic, Environmental, and Social Optimization: A Comprehensive AHP-DEA Approach from the Waste Transportation Sector"

_sustainability, doi:10.3390/su152115372_

Round 1

Reviewer 1 Report

Please review the instructions for authors, specifically in the style of references and how to cite in the document.

Reviewer 2 Report

The article entitled "Achieving a sustainable transportation system through economic environmental, and social optimization: a comprehensive AHP-DEA approach from waste transportation sector" is a scholarly work.  Although the paper addresses a very interesting topic there are a few comments and also improvement must be done to increase its quality and make usefull to comparison with work from other researchs.   The reviewer has the following observations:   • On page 3 (Lines 125 to 127), to offer a complete context, it is necessary to provide more characteristics of the city where the study is conducted, such as its size, population, and population density.   • On page 3 (Lines 131 to 136), although the waste collection system is briefly mentioned, it is necessary to describe its characteristics in more detail: the type of fleet used (electric trucks, gasoline, etc.), whether the collection is done house by house or at fixed collection points, if there are specific schedules and days, what type of waste (organic, inorganic, both) is collected on each collection day, whether the collection of domestic and commercial waste is separate, and if the truck has any special features for organic waste collection.  Additionally, it would be important, for each of the 6 zones where waste collection services are provided, to know the total number of residents served and the population density (people per square meter).   • Regarding the evaluated routes in Table 5 (Page 16), the factors evaluated within each route are not clarified, such as topography and the overall condition of the streets, their width, road type, traffic directions, the area's mobility, traffic hours, whether it's a commercial, residential, tourist, or mixed-use area. The specification of where the data comes from (historical data, experience-based, or calculated using specific parameters) for modeling each of the evaluated routes is also unclear. What is the variation between the routes? Is it the order of collection points, the collection schedule, or what characteristics were used to model each evaluated route? Regarding the efficiency results, what do the most efficient routes have in common?   • On page 18 (Lines 661 to 670), it is mentioned that similar results were obtained to other research in the area. However, the studies cited in the document emphasize that promoting the use of low-carbon and energy-efficient transportation minimizes negative impacts and provides greater environmental benefits to the route system. Please explain similarities.   • On page 20 (Lines 760 to 761), it is mentioned that the study provides a vision of the potential benefit of integrating sustainability into decision-making. However, there is no valuation of the inclusion of this aspect. In other words, there is no comparison between a route selection "without considering sustainability" and a route selection "considering sustainability" to demonstrate that the decision would have been different.    

Reviewer 3 Report

The authors proposed a hybrid multi-criteria decision-making (MCDM) approach for sustainable waste transportation, combining analytic hierarchy process (AHP) and data envelopment analysis (DEA) to enable an efficient route selection, balancing conflicting requirements and diverse perspectives. However,the approach empowers for the decision-makers and policymakers how to develop effective tools for waste transportation in developing countries, contributing to the growing body of research on sustainable waste management practices.

Average

Reviewer 4 Report

In this paper a hybrid multi-criteria decision-making approach combining AHP and DEA for route selection is suggested. I think that the suggested method has novelty and it contributes to the literature. However in this paper, Medium-sized cities are mentioned. According to what? This needs to be explaned. In result section, what is the output? It should be explaned. Which scenarios? Why only zone A?
